# Biosynthesized Highly Stable Au/C Nanodots: Ideal Probes for the Selective and Sensitive Detection of Hg^2+^ Ions

**DOI:** 10.3390/nano9020245

**Published:** 2019-02-12

**Authors:** Sada Venkateswarlu, Saravanan Govindaraju, Roopkumar Sangubotla, Jongsung Kim, Min-Ho Lee, Kyusik Yun

**Affiliations:** 1Department of Nanochemistry, Gachon University, Gyeonggi-do 13120, Korea; venkisada67@gmail.com; 2Department of Bionanotechnology, Gachon University, Gyeonggi-do 13120, Korea; biovijaysaran@gmail.com; 3School of Integrative Engineering, Chung-Ang University, Seoul 06974, Korea; 4Department of Chemical and Biological Engineering, Gachon University, 1342 Seongnam Daero, Seongnam-Si, Gyeonggi-do 13120, Korea; gachonroop@gmail.com (R.S.); jongkim@gachon.ac.kr (J.K.)

**Keywords:** onion leaves, GCNDs, orange fluorescent, TEM, mercury ion sensor

## Abstract

The enormous ongoing industrial development has caused serious water pollution which has become a major crisis, particularly in developing countries. Among the various water pollutants, non-biodegradable heavy metal ions are the most prevalent. Thus, trace-level detection of these metal ions using a simple technique is essential. To address this issue, we have developed a fluorescent probe of Au/C nanodots (GCNDs-gold carbon nanodots) using an eco-friendly method based on an extract from waste onion leaves (*Allium cepa-red onions*). The leaves are rich in many flavonoids, playing a vital role in the formation of GCNDs. Transmission electron microscopy (TEM) and Scanning transmission electron microscopy-Energy-dispersive X-ray spectroscopy (STEM-EDS) elemental mapping clearly indicated that the newly synthesized materials are approximately 2 nm in size. The resulting GCNDs exhibited a strong orange fluorescence with excitation at 380 nm and emission at 610 nm. The GCNDs were applied as a fluorescent probe for the detection of Hg^2+^ ions. They can detect ultra-trace concentrations of Hg^2+^ with a detection limit of 1.3 nM. The X-ray photoelectron spectroscopy results facilitated the identification of a clear detection mechanism. We also used the new probe on a real river water sample. The newly developed sensor is highly stable with a strong fluorescent property and can be used for various applications such as in catalysis and biomedicine.

## 1. Introduction

Heavy metal ions have caused wide spread water pollution which has become a serious threat to living organisms including humans [1,2]. These heavy metal ions (lead, arsenic, chromium, cadmium, and mercury) mainly enter water systems from various industries [3,4]. Among these pollutants, mercury is one of the most hazardous and is mainly released from electroplating, battery, and coal industries as well as from medical waste and chlor-alkali plants [5,6]. This toxic metal poses health risks which include damage to DNA, the brain, red blood cells, kidneys, and the liver and it is known to compromise the immune system [7,8]. The United States Environmental Protection Agency has set the allowable limit of Hg^2+^ in drinking water to less than 0.002 mg/L [9]. Thus, there is an ever-growing demand for sensitive and selective detection of Hg^2+^ in soil, food, air, and water [10,11]. Various techniques have been developed for the detection of Hg^2+^ ion, including electrochemical and colorimetric sensors [12,13]. However, most new techniques are limited in their applications because they have high implementation costs, are difficult to handle, require long sample preparation time, and exhibit low sensitivity and selectivity. Therefore, there is an urgent need to develop a simple method for fast detection of Hg^2+^. Recently, various fluorescent probes for the detection of Hg^2+^ ions based on carbon dots (CDs), organic polymers, organic dyes, metal fluorescent nanoparticles, and metal organic frames were reported [14,15,16,17,18].

Among these materials, CDs have unique properties such as strong luminescence, biocompatibility, and high stability. Therefore, CDs have gained considerable research attention for use in biomolecular and heavy metal ion sensing as well as other biological applications [19,20,21]. In addition, newly developed materials based on the combination of noble metal nanoparticles (Au and Ag) and CDs possess enhanced optical, photoluminescent, electrochemical, and catalytic properties [22,23,24]. Specifically, gold clusters can be ultra-small (<2 nm) and exhibit strong fluorescence, quantized charging, magnetism with good biocompatibility, eco-friendliness, and excellent photostability [25,26,27]. The oscillation of highly-concentrated free electrons present in gold clusters can considerably enhance the fluorescent property by the interaction between the metallic surface and the host CDs [28,29,30]. Based on the fluorescent property of gold nanoparticles, the creation of several sensors (Au/Ag, Au/Cu, and dye-encapsulated Au nanoparticles) for the detection of Hg^2+^ ions has been reported [31,32,33]. The reported methods have some disadvantages such as having bimetallic state and toxic dyes which enhance the cost and cause secondary pollution. These disadvantages can be overcome by a careful combination of CDs and Au nanoparticles [8]. Recently, Liu et al. and He et al. reported CDs Au-nanocluster hybrid materials applied as a fluorescent probe for the detection of Hg^2+^ ions [34,35]. However, these materials were synthesized using expensive, commercially available molecules (l-cysteine, diethylenetriamine, etc.) as the carbon precursors. To overcome these drawbacks, a simple method that uses freely available natural waste materials is required.

Here, we describe the synthesis of uniformly dispersed strongly fluorescent GCNDs using waste onion leaves (*Allium cepa-red onions*) according to a microwave-assisted method. The annual onion production in South Korea is nearly 1,700,000 metric tons. Onions are the second most cultivated vegetable crop in the world. These are an abundant source of many phytonutrients and phenolic compounds and have excellent antioxidant properties and can protect against various pathologies like cardiovascular and neurological diseases [36]. Various types of onions (white, red, etc.) are available in the world throughout the year; among them, red onions are rich in many flavonoids (quercetin, kaempferol, etc.), polysaccharides (ketose, fructofuranosylnystose, glycoside), anthocyanins, gallic acid, chlorogenic acid, p-coumaric acid, sinapic acid, and other sulfur (thiosulfinates) components etc. [37,38]. With many of these compounds having oxygen, nitrogen functional groups are attached to an aromatic and aliphatic carbon skeleton [39]. These biomolecules play a vital role in the formation of nanoparticles by reducing the metallic ions [40,41]. Moreover, these onion leaves have several applications in various fields like capacitor etc. [42]. This provides a huge precursor source for the synthesis of carbon dots which can also act as a reducing agent for the formation of gold nanoparticles. The synthesized GCNDs are considerably small (~2 nm) and give a strong orange fluorescence. Moreover, the biogenic GCNDs are stable for more than seven months. These GCNDs show excellent selectivity and sensitivity towards Hg^2+^ ions and the mechanisms involved are clearly explained using X-ray photoelectron spectroscopy (XPS) and Fourier-transform infrared spectroscopy (FTIR) techniques. Moreover, these GCNDs may also be useful for biomedical and catalytic applications. Finally, we have performed the reproducibility of the onion leaves extract and GCNDs using several red onions that were cultivated in different areas.

## 2. Materials and Methods

### 2.1. Materials

Gold (III) chloride trihydrate (HAuCl_4_·3H_2_O), l-glutathione (GSH), and mercury nitrate monohydrate (Hg (NO_3_)_2_·H_2_O) were purchased from Sigma-Aldrich (Gyeonggi-do, Seoul, South Korea). The waste red onion leaves were collected from a local market in Seongnam, South Korea. Deionized water was used throughout the experiments.

### 2.2. Synthesis of GCNDs

The GCNDs were synthesized by an extract from red onion leaves. The collected onion leaves were washed with de-ionized water and cut into small pieces. Afterward, 15 g of pieces were placed into a 250 mL round bottom flask containing 100 mL of DI water and were refluxed for 1 h at 85 °C. The resulting solution was cooled and filtered with cheese cloth. The filtrate was stored for further experiments. A typically and freshly prepared 5 mL of 30 mM of HAuCl_4_·3H_2_O aqueous solution and 1 mL of 75 mM GSH were mixed. The mixture was kept at 80 °C for 1 h, subsequently 20 mL of onion leaf extract was added, and the mixture was placed in a microwave oven for 10 min. A pale brown color was obtained by filtration of the reaction mixture. Subsequently, the mixture was dialyzed with a cellulose ester membrane (molecular weight cut-off: 2 kDa) against DI water for 24 h. Subsequently, powder was obtained by a freeze-drying process. The final product was used for further experiments. Scheme 1 shows an illustration of the formation of GCNDs. Onion extract acts as a reducing agent as well as carbon precursor which will initiate the growth of GCNDs. Moreover, the gold colloidal suspension which contains GSH molecules act as a stabilizing agent to the gold nanodots formation. Finally, the stability of the GCNDs depends on the carbon template which is internally conglomerate with the gold suspension and form stable GCNDs. The tiny gold nanoclusters were homogeneously distributed to the carbon skeleton.

### 2.3. Characterization

Diluted GCNDs samples were examined in dark field images using a PTI UV illuminator (Horiba scientific, Piscataway, NJ, USA) at 365 nm. UV-Vis spectra were measured (Varian Cary 100 UV-Vis spectrophotometer, Palo Alto, CA, USA). The photoluminescence (PL) intensity was recorded by QuantaMaster (Photon Technology International, Birmingham, NJ, USA) which was equipped with a xenon lamp (Arc Lamp Housing, A-1010B™), monochromator and power supply (Brytexbox, NJ, USA). The size and morphological characterization were determined using a JEOL JEM-ARM 200F series TEM instrument (Jeol, Peabody, MA, USA). Atomic force microscopy (AFM) (JPK NanoWizard II bio-atomic force microscope, Berlin, Germany) was performed with a JPK NanoWizard II bioatomic force microscope in contact mode to determine the surface morphology. X-ray diffraction (XRD) patterns of the particles were obtained using a Rigaku Rint 2200 Series X-ray Automatic Diffractometer (Cu Kα radiation at a wavelength of 1.5406 Å) (Rigaku Corp., The Woodlands, TX, USA). The GCNDs functional groups were illustrated using FT-IR with a Thermo Nicolet iS-10 spectrometer (BRUKER FT-IR Vertex 70, Billerica, MA, USA) using a KBr pellet in transmission mode. The detailed composition of the GCNDs was analyzed using a Thermo Scientific (Quanta Master, Photon Technology International, Birmingham, NJ, USA) K-Alpha^TM+^ (XPS) system equipped with 100–4000 eV range of motion, 180° double focusing hemispherical analyzer (Thermos scientific, Oklahoma City, OK, USA) with 128-channel detector (Thermos scientific, Oklahoma City, OK, USA), and Al Ka micro-focused X-ray source.

### 2.4. Detection of Hg^2+^ Ions Using Fluorescent GCNDs Probe

For the determination of Hg^2+^ ions at room temperature, various concentrations of Hg^2+^ (0–70 µM) solution were prepared. A 100 µL portion of the prepared Hg^2+^ solution was added to a cuvette containing 100 µL of GCNDs; afterward, a certain volume of PBS (0.1 M, pH 7.5) buffer solution was added to make up a total volume of 2 mL. The reaction mixture was incubated for 5 min and the PL of the sample was measured with excitation at 380 nm. For investigating the high selectivity of Hg^2+^, a similar method was applied, however with various interference ions.

## 3. Results and Discussion

### 3.1. Material Characterization 

Figure 1a,b show the High-resolution transmission electron microscopy (HRTEM)images of the GCNDs. They are approximately 2 nm and are uniformly distributed within clusters. In addition, Figure 1c–f shows the STEM-EDS elemental mapping which clearly indicates the presence of elemental C, O, and Au. The STEM image in Figure 1c clearly indicates the Au/C alloy-type nanoparticles which means that CDs are strongly integrated with the Au clusters. The phase purity is confirmed by the XRD analysis. Figure 1h illustrates the XRD pattern with the peaks that appear at 16.9° and 22.0° with the plane of (002) corresponding to amorphous carbon and a partially graphitized phase for the CDs. The peaks located at 2θ values of 38.2° and 44.1° refer to the Au present in the form of clusters. Further details of the morphology are shown in the AFM image in Figure 2a which shows uniformly distributed clusters mixed with many tiny GCNDs. The latter are clearly shown in the 3D AFM image in Figure 2b. Moreover, the AFM height profile clearly indicates that the height of the GCNDs is <4.1 nm. The surface functional groups and composition of GCNDs were analyzed using XPS and FT-IR. The survey scan spectra of GCNDs in Figure 3a(i) show five peaks at 84.2, 163.8, 284, 400, and 530 eV attributed to Au-4f, S-2p, C-1s, N-1s, and O-1s, respectively. After reacting with Hg^2+^ ions in Figure 3a(ii), the intensities of C-1s and O-1s peaks slightly decreased and S-2p, N-1s were shifted to a lower binding energy region, which is clearly shown in Figure 3d(ii),f(ii). Moreover, a new peak appeared at 105 eV, which is attributed to the presence of Hg^2+^, as shown in Figure 3a(ii),g. The high-resolution C-1s spectra of the GCNDs reveal three deconvoluted main peaks at 283.58, 289.38, and 291.78 eV which indicate the presence of C-O, C=O, and C-C species, respectively, as shown in Figure 3b. In contrast, the high-resolution O-1s spectrum in Figure 3c reveals two main peaks at 531.08 and 538.58 eV, representing C-O and C=O, respectively. Figure 3d shows the high-resolution S-2p spectrum with a characteristic peak at 164.5 eV, which is attributed to S-2p-3/2 of -SH groups. Moreover, the high-resolution spectrum of N-1s shows two peaks at 400.5 and 401.8 eV, which is attributed to the presence of primary and secondary amine groups, as shown in Figure 3f. A high-resolution spectrum of gold shows peaks at 84.38 and 86.58 eV, representing Au-4f-5/2 and Au-4f-7/2, as shown in Figure 3e [35]. Furthermore, the surface functional groups of the onion leaves extract were analyzed by using FT-IR. Figure 4(i) illustrates a broad peak around 3580 cm^−1^ which indicates the presence of hydroxyl groups (-OH), and the peak at 2986 cm^−1^ belongs to the -CH moiety. Moreover, for the peaks at 1722, 1507, 1426, 1297, 1160, 1065, and 830 cm^−1^, which are ascribed to the carbonyl (-C=O) group of acids and keto, there are C=C, -C-H, -C-N, -C-O, and -C-S groups of biomolecules (flavonoids, polysaccharides (ketose), anthocyanins, gallic acid, sinapic acid, and thiosulfinates) [43,44,45]. However, after the formation of GCNDs, the leafy extract peaks almost shifted, as shown in Figure 4(ii) where the peaks appeared at 3513, 3297, 2528, 1753, 1625, and 788 cm^−1^, which is attributed to the stretching vibrations of -OH, -NH_2_/-NH, -SH, C=O, amide, and C-S groups, respectively. After treating with Hg^2+^ ions in Figure 4(iii), the FT-IR peaks at 1753, 1160 are shifted to 1708, 1146 cm^−1^, and the peak at 2528 cm^−1^ almost disappears, which confirms that -C=O, -C-N, and -SH groups are strongly attracted to the target Hg^2+^ ions. This result indicates the successful formation of leaves extract mediated GSH functionalized GCNDs.

### 3.2. Optical Properties of GCNDs

The UV-Vis absorbance spectra of onion leaves extract are shown in Figure 5a. The spectrum shows five bands at 265, 293, 320, 342, and 377 nm, which is attributed to characteristic peaks of biomolecules like anthocyanins, quercetin, and kaempferol etc., which are present in the extract and can act as a reducing agent as well as a carbon precursor in the formation of GCNDs. The results are well matched with reported literature [46,47,48]. Figure 5b,c show the UV-Vis absorbance and PL spectra of the synthesized GCNDs. The UV-Vis absorbance bands at approximately 290 and 340 nm (attributed to the presence of π–π* and n–π*, respectively) are attributed to C=C bonds of the sp^2^ aromatic moiety of the CDs and to C=O, COOH and -NH bonds (Figure 5b) [49]. This result indicates that the GSH groups are chelated on the surface of GCNDs, as shown in Scheme 2. Moreover, the inset in Figure 5b clearly shows the orange fluorescent behavior of GCNDs in the absence and presence of UV light. Figure 5c shows that the PL spectra of GCNDs has an emission at 610 nm when excited at 380 nm. In addition, Figure 5d(i–iii) shows the PL spectra of bare CDs, pure Au clusters, and GCNDs, which have emission peaks at 430, 645, and 610 nm, respectively. The GCNDs are given a blue shift owing to the presence of CD moiety. The fluorescent GCNDs show a quantum yield of 11%. In Figure 6, we show the optimized stability of GCNDs under different conditions. Figure 6a shows that the various excitation (from 360 to 460 nm) is dependent on emission peaks. The PL excitation dependent emission spectrum indicates that the PL emission peak exhibited a blue shift transition from 630 to 530 nm when the excitation wave lengths were from 360 to 460 nm. This shift is mainly attributed due to the presence of carbogenic core moiety. Figure 6b shows the pH-dependent fluorescence behavior of GCNDs. The PL spectra are recorded at various pH values (2.0, 3.0, 5.0, 6.0, 7.5, 9.0, 10, and 12). The PL intensity is low at lower pH values (quenched by as much as 59%), however at higher values, PL intensity looks almost the same for pH values higher than 7. At lower pH values, there is competition between protons (H^+^) and Hg^2+^ ions which will lower the probe sensitivity. In contrast, at higher pH values, the Hg^2+^ ions can precipitate to Hg(OH)^+^, Hg(OH)_2_, and Hg(OH)_3_^−^, which will reduce the sensitivity of the probe [9]. In addition, the stability of GCNDs after various time intervals at room temperature was investigated. As shown in Figure 6c, the PL intensity for 300 days is nearly 92%. The stability of the GCNDs depends on the carbon template which is internally conglomerate with the gold suspension which form stable GCNDs. Owing to the excellent stability of these GCNDs, they appear useful for various biological and catalytic applications.

### 3.3. Sensitivity and Selectivity of GCNDs Towards Hg^2+^ Ions

The as-synthesized GCNDs were used to determine their effectiveness in the detection of Hg^2+^. The fluorescence sensor was tested using different concentrations of Hg^2+^ (0–70 µM), as shown in Figure 7a. The PL quenching of GCNDs in the presence of Hg^2+^ was 95% upon the addition of Hg^2+^ ions. The relative changes of fluorescence intensity from the initial intensity (F_0_/F) at 380 nm, with Hg^2+^ at 0–70 µM, are shown Figure 7b. Depending on the concentration of Hg^2+^, the fluorescence intensity is gradually quenched and the detection limit is achieved up to 1.3 nM, which is much better than the other methods listed in Table 1 [50,51,52,53,54,55]. This is due to the presence of -SH and amine groups on the GCNDs surface that can react strongly with Hg^2+^ ions. The limit of detection was determined by using the following equation:LOD=3×σ/m
where σ is the standard deviation of the lowest tested concentration and m is the slope [14]. Based on the XPS and FT-IR results, we were able to identify a clear mechanism. Moreover, the Au nanoclusters and carbon skeleton that form each GCNDs support the functions of the fluorescence resonance energy transfer (FRET) sensor [55]. To determine their selectivity, we investigated the effect of interference ions such as Fe^2+^, Fe^3+^, Ni^2+^, Ag^+^, Cd^2+^, Mn^2+^, Cr^2+^, Ca^2+^, and Mg^2+^ with concentrations that are two-fold higher than that of the target Hg^2+^ (70 µM) ions. The fluorescence intensity of the GCNDs was quenched 95% by the Hg^2+^ ions, whereas the other metal ions did not show any significant fluorescence quenching, as shown in Figure 7c. Along with Hg^2+^, other soft acid metal ions such as Ag^+^ and Cd^2+^ were also studied. It is important to note that our probe could distinguish Hg^2+^ from Ag^+^ and Cd^2+^. This may be that the 5d series of Hg^2+^ can form a stable amalgam compared to the 4d series of Ag^+^ and Cd^2+^, which is in agreement with the literature [1,56]. In addition, we used the new sensor to detect Hg^2+^ ions in a river water sample which was collected from the Han River (Seoul, South Korea). The collected water was filtered using a 0.45 µm membrane to remove impurities and was centrifuged at 20,000 rpm for 30 min before analysis. For the pre-treated river water, various concentrations of Hg^2+^ ions (0–40 µM) and various interference ions (100 µM) were added. Figure 8a shows the PL quenching of GCNDs in the presence of Hg^2+^ ions, and Figure 8a(i,ii) inset photos indicate the images of GCNDs before and after Hg^2+^ addition under UV illumination. The detection limit is 126 nM. Figure 8b shows the selectivity of GCNDs. The river water contains various sources of interference, such as halogen ions, carbonates, bicarbonates, phosphates, and some organic dye molecules. The fluorescence intensity of GCNDs in real water is quenched by as much as 80% of Hg^2+^ ions. These results reveal that the newly developed GCNDs probe is useful for practical environmental applications.

### 3.4. Reproducibility Study

For confirmation of the reproducibility of the material synthesis and its application, we have collected the red onion leaves from various places (Daegu and Incheon, South Korea) and the extract was prepared using the same method. The prepared extract was characterized by using FT-IR and UV-Vis absorption spectroscopic techniques. Appendix A shows that the FT-IR spectra of onion leaves extract for both samples exhibit the major peaks at ~3500, ~1700, and ~1070 cm^−1^, which is attributed to the hydroxyl (-OH), carbonyl (-C=O), and ether (-C-O), and similar groups are presented in both samples. The functional groups may have derived from the backbone of the following molecules; flavonoids, polysaccharides, anthocyanins, gallic acid, etc. that can play a key role in the formation of nanoparticles. In addition, the UV-Vis absorbance spectra of these onion leaf extracts are shown in Appendix A. The absorption curve is similar to the spectra in Figure 5a, indicating that the extract may have a composition of biomolecules like anthocyanins, quercetin, and kaempferol etc. The presented moieties can act as a reducing agent as well as a carbon precursor in the formation of GCNDs. Moreover, Appendix A shows the UV-Vis absorption spectra of the synthesized GCNDs. The UV-Vis absorbance bands are approximately similar to the bands that are shown in (Figure 5b). The inset in Appendix A clearly shows the orange fluorescent behavior of GCNDs in the absence and presence of UV light. Finally, we have observed quenching behavior of the re synthesized GCNDs towards the target Hg^2+^ ion; the result is shown in Appendix A. These results confirm the regeneration of the extract as well as synthesized material by red onion species collected from various cultivated areas. This result indicates that the red onion cultivated in different places also shows similar results and therefore, the red onion extract can have reproducibility for this application.

### 3.5. Quenching Mechanism of GCNDs by Hg^2+^ Ions

According to Pearson’s HSAB theory, soft acid groups can rapidly react with Hg^2+^ ions. Therefore, the high selectivity is attributed to the presence of soft base (-SH), which is highly reactive towards soft acid (in this case, Hg^2+^). Moreover, after treatment of Hg^2+^ ions, the high-resolution XPS spectra of S-2p and N-1s reveal that peaks from 164.5 and 400.5 eV shifted to 399.2 and 163.9 eV, as shown in Figure 3d(ii),f(ii). This is due to electron transfer between -SH and Hg^2+^ ions, as shown in Scheme 2, in which case the PL intensity is quenched. A high-resolution spectrum of mercury shows peaks at 100.68 and 104.48 eV, which represent Hg-4f-7/2 and Hg-4f-5/2, respectively (Figure 3g). In addition, the FT-IR peaks at 1753, 1160 are shifted to 1708, 1146 cm^−1^, and the peak at 2528 cm^−1^ almost disappears, which confirms that -C=O, -C-N, and -SH groups are strongly attracted to the target Hg^2+^ ions that are shown in Figure 4(iii). This result concurs with those reported in the literature [57].

## 4. Conclusions

In this work, we developed an eco-friendly method for the synthesis of Au/C nanodots (GCNDs) using waste red onion leaves extract. Onions are the second most cultivated vegetable crop in the world. These are an abundant source of many flavonoids and phenolic compounds which can act as a reducing, stabilizing agent and carbon precursor. The synthesized GCNDs showed a uniform particle size of ~2 nm, as confirmed by HRTEM, STEM-EDS elemental mapping and AFM analysis. The surface functional groups were identified from FT-IR and XPS results. The GCNDs showed a strong orange fluorescence in the UV-Visible spectrometry. The developed GCNDs were applied as a fluorescent probe for the detection of toxic Hg^2+^ ions in water. Owing to the carbon precursor, the GCNDs were highly stable up to 300 days and showed an excellent detection limit of Hg^2+^ ions (as low as 1.3 nM). This detection limit is one of the lowest among the reported materials. Owing to the -SH functional groups, the GCNDs showed high selectivity towards Hg^2+^ ions, even in the presence of interference. The detailed mechanism is clearly explained using XPS and FT-IR techniques. Moreover, the newly developed GCNDs were used successfully to test a real river water sample and they provided a detection limit of 126 nM. We have also regenerated the extract as well as synthesized material by red onion leaves that were collected from various cultivated areas. These newly developed GCNDs may also be useful for other applications such as catalysis, electrochemistry, and biomedicine.

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
