# Peer review of "Biosynthesized Highly Stable Au/C Nanodots: Ideal Probes for the Selective and Sensitive Detection of Hg2+ Ions"

_nanomaterials, 2019, doi:10.3390/nano9020245_

Round 1

Reviewer 1 Report

the paper is fine now and can be published

Author Response

Comments and Suggestions for Authors

The paper is fine now and can be published

Thank you for the comment 

Reviewer 2 Report

The two problems detected in the first version of the paper were not answered. 

-      Indeed, as the authors now describe, the chemical composition of the onion reflux (and there are several different types of onions) is quite complex and must be qualified and quantified (at least the main constituents) in order to allow the reproduction of this results in other laboratories. Moreover, from the scientific point of view, it would be important to know the molecules that confer the observed reactivity to the synthesized CD. 

-      The importance of the gold nanoparticles for the performance of the sensor is not clear because there are descriptions of CD with better performance than the proposed CD+Au NPs system. 

Author Response

Comments and Suggestions for Authors

The two problems detected in the first version of the paper were not answered.

-      Indeed, as the authors now describe, the chemical composition of the onion reflux (and there are several different types of onions) is quite complex and must be qualified and quantified (at least the main constituents) in order to allow the reproduction of this results in other laboratories. Moreover, from the scientific point of view, it would be important to know the molecules that confer the observed reactivity to the synthesized CD.

We appreciate the reviewer’s comment. The reviewer first raises an issue about reproducibility and we successfully demonstrated reproducibility in the revised manuscript with common chemical moieties presented in onion extract. We explained the role of the common moieties in the extract including various flavonoids, polysaccharides, anthocyanins, gallic acid etc., which was qualified by UV-Vis and FT-IR analysis. Further analysis of constituent of the extract is beyond the scope of this manuscript. We hope the presented reproducibility and the qualitative analysis result is satisfactory for the reviewer.

-      The importance of the gold nanoparticles for the performance of the sensor is not clear because there are descriptions of CD with better performance than the proposed CD+Au NPs system.

The carbon dots with blue fluorescence has been reported by many others. In our work, however, the GCNDs showed a strong orange fluorescence that is hardly demonstrated in pure CD materials. In addition, this composite showed higher stability than pure carbon doe because pure CD does not show long stability. For that reason, we used gold nanoparticles in this work to develop GCNDs probe.

This manuscript is a resubmission of an earlier submission. The following is a list of the peer review reports and author responses from that submission.

Round 1

Reviewer 1 Report

The English language used has to be checked once more because there are several mistakes in the language

for example line 191: replace the word materials with : methods

Table 1: instead of :  Comparison of GCNDs for Hg2+ ions various reported materials

it should be Comparison of the present method with other various methods reported in the literature 

and so on

The method is very good and the authors did an excellen work on the characterization

However, I would have preferred if the authors have detected a different species than mercury

They would for example solve this problem by writing that the mercury determination is an example of an application!!!

Author Response

Manuscript ID: nanomaterials-421542
Type of manuscript: Article
Title: Biosynthesized Highly Stable Au/C Nanodots: Ideal Probes for Selective and Sensitive Detection of Hg2+ Ions
Authors: Sada Venkateswarlu, saravanan govindaraju, Roopkumar Sangubotla,  Jongsung Kim, kyusik yun *

Reviewer 1

Open Review

English language and style

(x) Extensive editing of English language and style required 
( ) Moderate English changes required 
( ) English language and style are fine/minor spell check required 
( ) I don't feel qualified to judge about the English language and style 

Yes

Can   be improved

Must   be improved

Not   applicable

Does   the introduction provide sufficient background and include all relevant   references?

(x)

(   )

(   )

(   )

Is   the research design appropriate?

(x)

(   )

(   )

(   )

Are   the methods adequately described?

(x)

(   )

(   )

(   )

Are   the results clearly presented?

(x)

(   )

(   )

(   )

Are   the conclusions supported by the results?

(x)

(   )

(   )

(   )

Comments and Suggestions for Authors

The English language used has to be checked once more because there are several mistakes in the language

Thank you for the suggestion, now we have polished the English language throughout the manuscript.

For example, line 191: replace the word materials with: methods

As per your suggestion, we have corrected the word.

Table 1: instead of:  Comparison of GCNDs for Hg2+ ions various reported materials             it should be Comparison of the present method with other various methods reported in the literature and so on

Thank for your suggestion, now we have corrected the sentence in the revised manuscript.

The method is very good, and the authors did an excellent work on the characterization

Thank you for the comment.

However, I would have preferred if the authors have detected a different species than mercury

Yes, we have analyzed various metal ions like Fe2+, Fe3+, Ni2+, Ag+, Cd2+, Mn2+, Cr2+, Ca2+, and Mg2+ shown in Figure 7c.

They would for example solve this problem by writing that the mercury determination is an example of an application!!!

Thank you for the comment, the further research based on this developed probe is in progress.

Reviewer 2 Report

Tha authors report the association of Au clusters with carbon dots produced from onion leaves and the use of these nanohybrids for the detection of Hg2+ in water. Some of the results presented may be of interest. The following comments should be considered by the authors before submitting the revised version :

- specify how the onion extract was produced from onion leaves.

- clarify paragraph 2.2. Scheme 1 is also misleading : the onion leaves extract was added to preformed Au clusters.

- line 128 : FT-IR

- indicate how GCNDs were isolated and purified. The presence of free SH functions in the nanohybrids is surprising as SH functions should strongly associate to Au clusters.

- paragraph 3.2 : the PL QY of GCNDs should be provided.

- lines 172-173 : clarify the sentence.

- it would be of interest for the readers to provide the optical properties of Au clusters, of the carbon dots and discuss in this context those of GCNDs.

- figure 6a : 1) The PL emission maximum depends on the excitation wavelength. The text must be modified. 2) Specify for each spectrum the excitation wavelength. The results should be discussed.

- The inset of figure 7a can only poorly be seen. From my opinion, the plot of F/F0 vs [Hg2+] should be provided as a full size figure. Indicate how the detection limit was determined.

- check the legend of the y axis in figures 7b and 8b. Results described in Figure 8 should be better discussed.

- thiol and amine functions should also interact with other transition metal cations. The discussion should be revised. To prove the mechanism, the authors should evaluate the quenching in the presence of Ag+ or Cd2+ cations.

- line 205 : the detection limit is 126 nm (?)

- Along the whole manuscript, results should be better discussed in the context of literature.

- the language must be improved and the manuscript carefully checked by the authors as it contains many typing errors.

Author Response

Manuscript ID: nanomaterials-421542
Type of manuscript: Article
Title: Biosynthesized Highly Stable Au/C Nanodots: Ideal Probes for Selective and Sensitive Detection of Hg2+ Ions
Authors: Sada Venkateswarlu, saravanan govindaraju, Roopkumar Sangubotla,  Jongsung Kim, kyusik yun *

Reviewer 2

English language and style

( ) Extensive editing of English language and style required 
(x) Moderate English changes required 
( ) English language and style are fine/minor spell check required 
( ) I don't feel qualified to judge about the English language and style 

Yes

Can be improved

Must be improved

Not applicable

Does the introduction provide sufficient   background and include all relevant references?

( )

( )

(x)

( )

Is the research design appropriate?

( )

( )

(x)

( )

Are the methods adequately described?

( )

( )

(x)

( )

Are the results clearly presented?

( )

(x)

( )

( )

Are the conclusions supported by the   results?

( )

( )

(x)

( )

Comments and Suggestions for Authors

The authors report the association of Au clusters with carbon dots produced from onion leaves and the use of these nanohybrids for the detection of Hg2+ in water. Some of the results presented may be of interest. The following comments should be considered by the authors before submitting the revised version:

Specify how the onion extract was produced from onion leaves.

Thank you for the comment, now we have given the detailed procedure of onion extract in the revised manuscript.

Clarify paragraph 2.2. Scheme 1 is also misleading: the onion leaves extract was added to preformed Au clusters.

We appreciate the suggestion, now we have corrected the paragraph with more details, and the scheme 1 also corrected in the revised manuscript.

line 128: FT-IR

Yes, now it is corrected

Indicate how GCNDs were isolated and purified. The presence of free SH functions in the nanohybrids is surprising as SH functions should strongly associate to Au clusters.

We appreciate the comment, now we have given the purification process in the revised manuscript in paragraph 2.2. 

Paragraph 3.2: the PL QY of GCNDs should be provided.

As per your suggestion, we have calculated the PL QY of fluorescent GCNDs and the calculated value is given in the revised manuscript.

lines 172-173: clarify the sentence.

Thank you for the comment, we have revised the lines 172-173 and polished the sentence.

It would be of interest for the readers to provide the optical properties of Au clusters, of the carbon dots and discuss in this context those of GCNDs.

As per your suggestion, now we have provided the optical properties of Au and bare carbon dots in the revised manuscript in paragraph 3.2. Figure 5c (i–iii) shows the PL spectra of bare CDs, pure Au clusters, and GCNDs, which have emission peaks at 430, 645, and 610 nm, respectively. The GCNDs are given a blue shift owing to the presence of CD moiety.

Figure 6a : 1) The PL emission maximum depends on the excitation wavelength. The text must be modified. 2) Specify for each spectrum the excitation wavelength. The results should be discussed.

As per reviewer suggestion, now we have included the text in Figure 6a. The results are discussed in a detailed manner in the revised manuscript.

The inset of figure 7a can only poorly be seen. From my opinion, the plot of F/F0 vs [Hg2+] should be provided as a full-size figure. Indicate how the detection limit was determined.

Thank you for the suggestion, now we have given a full-size Figure 7b for the linear fit of quenched fluorescence intensity as a function of Hg2+ concentration. We have given the equation for the detection limit in the revised manuscript.  

Check the legend of the y axis in figures 7b and 8b. Results described in Figure 8 should be better discussed.

Thank you for the comment, now have corrected the Figure 7b and 8b. In the revised manuscript the Figure 7b is moved to Figure 7c. The results of Figure 8 is discussed in the revised manuscript.

Thiol and amine functions should also interact with other transition metal cations. The discussion should be revised. To prove the mechanism, the authors should evaluate the quenching in the presence of Ag+ or Cd2+ cations.

Yes, thiol and amine functions groups may also interact with other transition metal cations. Therefore, we have studied the effect of coexisting ions including Ag+ or Cd2+ cations.

line 205: the detection limit is 126 nm (?)

Now we have given correct unit 126 nM.

Along the whole manuscript, results should be better discussed in the context of literature.

Thank you for the suggestion, now we have revised the result and discussion and given good emphasis.

The language must be improved, and the manuscript carefully checked by the authors as it contains many typing errors.

Thank you for the suggestion, now we have polished the English language throughout the manuscript and eliminated the typing errors.

Reviewer 3 Report

This paper describes the synthesis of a composite constituted by gold nanoparticles and carbon dots. The composite as obtained from the liquid resulting from refluxing onion leaves in water. The use of onion leaves without a detailed characterization of the liquid resulting from its refluxing is a critical point of the work. It raises two problems: (i) it is difficult for another research to reproduce this work because it would be difficult to obtain the same species of onion under the same cultivated and maturation conditions; (ii) without knowing in detailed the chemical composition of the liquid resulting from the reflux it is not possible to justify the obtained results. Another question about this work is the reason on the use of the gold nanoparticles to develop a mercury sensor. Indeed, there are many previous work using single carbon dots with similar or better sensing characteristics.

Round 2

Reviewer 2 Report

The authors have improved the manuscript but the following points must be considered before submitting a revised version.

- paragraph 2.2 : 1) indicate the mass of onion leaves used. 2) line 89 : correct into "the mixture was kept at 80°C for 1h"

- lines 94-95 : clarify the sentence. "carbon as the main element acting as quantum dots" ?

- lines 147-148 : correct into " GCNDs have been successfully functionalized with GSH"

- lines 175-177 : GCNDs do not emit at 610 nm when the excitation wavelength is tuned from 360 to 460 nm. Please correct the text.

- a major key point not clarified by the authors is the presence of free SH groups at the periphery of GCNDs. GSH is well-known to be highly sensitive to oxidation (for example, O2) and readily oxidized into GSSG. How can GCNDs be stable for 300 days ? Moreover, why do these SH functions not bound with Au atoms ?

- clarify the legend of figures 2 and 5

Author Response

Comments and Suggestions for Authors

The authors have improved the manuscript but the following points must be considered before submitting a revised version.

Paragraph 2.2 : 1) indicate the mass of onion leaves used. 2) line 89 : correct into "the mixture was kept at 80°C for 1h"

Thank you for the comment, now we have given the mass of onion leaves (15g) and also incorporated the sentence “the mixture was kept at 80°C for 1h" in the revised manuscript.

Lines 94-95 : clarify the sentence. "carbon as the main element acting as quantum dots" ?

As per your suggestion, the sentence was corrected.

Lines 147-148 : correct into " GCNDs have been successfully functionalized with GSH"

Thank you for the suggestion, the sentence " GCNDs have been successfully functionalized with GSH" incorporated in the revised manuscript.

Lines 175-177 : GCNDs do not emit at 610 nm when the excitation wavelength is tuned from 360 to 460 nm. Please correct the text.

As per your suggestion, we have corrected the lines 175-177: Figure 6a shows that the various excitation (from 360 to 460 nm) dependent emission peaks. Moreover, the excitation at 380 nm shows the high fluorescent emission (610 nm).

A major key point not clarified by the authors is the presence of free SH groups at the periphery of GCNDs. GSH is well-known to be highly sensitive to oxidation (for example, O2) and readily oxidized into GSSG. How can GCNDs be stable for 300 days? Moreover, why do these SH functions not bound with Au atoms?

We appreciate the comment; The gold colloidal suspension contains GSH molecules act as a stabilizing agent to the gold nanodots formation. Moreover, the gold nanodots are internally conglomerate with the carbon templet, which leads stability of GCNDs for 300 days. The GSH groups are chelated with Au atoms shown in the revised scheme 2. Owing to the lone pair electrons of -SH also form A bond with Hg2+ ions as per HSAB theory.

Clarify the legend of figures 2 and 5

Thank you for the comment, now we have given corrected the legend of Figures 2 and 5.
